Applications of dry chain technology to maintain high seed viability in tropical climates

Guzzon Filippo f.guzzon@cgiar.org 1
Costich Denise E. 2
Afzal Irfan 3
Barboza Barquero Luis 4
Monge Vargas Andrés Antonio 4
Vargas Ramírez Ester 4
Bello Pedro 5
Dahal Peetambar 5
Sánchez Cano César 6
Zavala Espinosa Cristian 6
Imran Shakeel 7
Patolo Soane 8
Tukia Tevita Ngaloafe 8
Van Asbrouck Johan 9
Nabubuniyaka-Young Elina 10
Gianella Maraeva 11
Bradford Kent J. 5
1 European Cooperative Programme for Plant Genetic Resources (ECPGR), c/o Alliance of Bioversity International, CIAT , Rome , Italy
2 Institute for Genomic Diversity, Cornell University , Ithaca , New York , United States of America
3 Seed Physiology Laboratory, Department of Agronomy, University of Agriculture , Faisalabad , Pakistan
4 Seed and Grain Research Center (CIGRAS) , San José , Costa Rica
5 Seed Biotechnology Center, Department of Plant Sciences, University of California , Davis , California , United States of America
6 International Maize and Wheat Improvement Center (CIMMYT) , Texcoco , Mexico State , Mexico
7 University of Agriculture, Faisalabad, UAF Sub-Campus Burewala , Faisalabad , Pakistan
8 MORDI TT (Mainstreaming of Rural Development Innovation Tonga Trust) , Nuku’alofa , Kingdom of Tonga
9 Rung Rueng Consulting Co., LTD/Rhino, Donmuang , Bangkok , Thailand
10 Centre for Pacific Crops and Trees (CePaCT), Land Resource Division (LRD), Pacific Community (SPC) , Suva , Fiji
11 Millennium Seed Bank, Royal Botanic Gardens Kew, Wakehurst Place , Ardingly , United Kingdom
Manjarrez Javier
Electronic publication date: 2024 Oct 11
Publication date: 2024
Volume: 12
Electronic Location ID: e18146
Received 2024 Feb 27; Accepted 2024 Aug 30
Copyright: ©2024 Guzzon et al.
Copyright year: 2024
Copyright holder: Guzzon et al.
License: This is an open access article distributed under the terms of the Creative Commons Attribution License, which permits unrestricted use, distribution, reproduction and adaptation in any medium and for any purpose provided that it is properly attributed. For attribution, the original author(s), title, publication source (PeerJ) and either DOI or URL of the article must be cited.
License URL: https://creativecommons.org/licenses/by/4.0/

Keywords: Community seed banks, Drying beads, Seed germination, Hermetic storage, Seed conservation, Seed longevity, Seed quality, Seed systems

Funding: USAID Costa Rica projects were financially supported by “Fundecooperación para el Desarrollo Sostenible” National Seed Office The Bangladesh projects were financially supported by USAID, the Costa Rica projects were financially supported by “Fundecooperación para el Desarrollo Sostenible” and the National Seed Office. The funders had no role in study design, data collection and analysis, decision to publish, or preparation of the manuscript.

==============================
Seed storage life in tropical areas is shortened by high humidity and temperature and the general inaccessibility to dehumidifying and refrigeration systems, resulting in rapid decreases in seed viability in storage as well as a high incidence of fungal and insect infestations. The dry chain, based on rapid and deep drying of seeds after harvest followed by packaging in moisture-proof containers, has been proposed as an effective method to maintain seed quality during medium-term storage in humid climates, even without refrigeration. In addition, seed drying with zeolite drying beads can be more effective and economical than sun or heated-air drying under these warm, humid conditions. In this paper, we review recent published literature regarding the dry chain, considering different crop species, storage environments and seed traits. In addition, we provide new original data on the application of dry chain methods and their implementation at larger scales in South Asia, Latin America and Pacific Island Countries. The clear conclusion is that the combination of reusable drying beads and waterproof storage containers enables the implementation of the dry chain in tropical climates, enhancing seed viability and quality in storage of many crop species. The dry chain approach can therefore significantly enhance seed security for farmers in many tropical countries. Finally, we propose actions and strategies that could guide further scaling-up implementation of this technology.

Introduction

Seed storage in tropical areas is particularly problematic due to high relative humidity (RH) that promotes insect and fungal infestations as well as seed ageing, manifested by rapid losses in seed quality and viability (Roberts & Ellis, 1989; Dickie et al., 1990; Pittendrigh et al., 2003). For most orthodox seeds, drying to low moisture content (MC; the relative amount of water in the seed based on either initial fresh weight or dry weight), and hermetic packaging to maintain it, can enable storage for up to several years without refrigeration with relatively small losses in viability (Ellis & Roberts, 1981; Ellis, 2022). Preventing these rapid losses in seed viability directly improves seed security for farmers and, as a result, food security for local communities. Seed conservation in some areas of low- and middle-income countries is also affected by the limited availability of dehumidifying and refrigeration systems that are recommended for the long-term storage of seeds (Hay & Timple, 2013; Guzzon et al., 2020).

Bradford et al. (2018) proposed the “dry chain”, i.e., “the initial dehydration of seeds to levels preventing fungal growth (RH < 65%) or insect activity (RH < 35%) followed by storage in moisture-proof containers”, as a solution for short- and medium-term seed storage (up to 5 years) in tropical and humid areas. In particular, drying with reactivable desiccants that can absorb water and bind it strongly can be a valuable option for seed drying in humid climates, as sun or heated-air drying are relatively ineffective under warm, humid conditions (Bradford et al., 2018). In the last decade, zeolite “drying beads” were tested in several tropical areas and found to be a very effective reusable desiccant for drying seeds to low equilibrium relative humidity. Equilibrium relative humidity (eRH) is the RH of the air around the seeds in a closed container (Hay, Rezaei & Buitink, 2022), its relationship with seed moisture content (SMC) at a given temperature is described by a moisture sorption isotherm (see Bewley et al., 2013). The seed drying stage must then be followed by storage in moisture-proof containers to maintain the low SMC achieved and prevent rehydration in humid climates. In recent years, several hermetic containers have been promoted to enhance quality in storage of dry grains and seeds in warm and humid areas (see e.g., Williams, Murdock & Baributsa, 2017).

The dry chain approach, understood as the initial desiccation of seeds to a safe MC, coupled with the use of zeolite beads as a desiccant and followed by storage in hermetic containers, has been tested and applied in several areas of Africa (Kenya, Tanzania), Asia (Bangladesh, India, Malaysia, Nepal, Pakistan, Philippines and Thailand), and Latin America (Costa Rica, Guatemala), and in different settings (i.e., community seed banks, genebanks, seed laboratories and seed companies). The dry chain was tested using different crop species, including amaranth (Amaranthus L.), cotton (Gossypium hirsutum L.), cucumber (Cucumis sativa L.), eggplant (Solanum melongena L.), groundnut (Arachis hypogaea L.), lablab bean (Lablab purpureus (L.) Sweet), maize (Zea mays L.), mung bean (Vigna radiata (L.) R.Wilczek), okra (Abelmoschus esculentus (L.) Moench), onion (Allium cepa L.), pea (Pisum sativum L.), pepper (Capsicum annuum L.), quinoa (Chenopodium quinoa Willd.), rice (Oryza sativa L.), sorghum (Sorghum Moench), soybean (Glycine max (L.) Merr.), tomato (Solanum lycopersicum L.), velvet bean (Mucuna pruriens (L.) DC.) and wheat (Triticum aestivum L.). In essentially all of these cases, use of the dry chain approach improved the maintenance of seed vigor and viability during storage compared to standard or traditional storage methods. These results indicate that the dry chain should be implemented globally, particularly in warm, humid climates, by farmers, conservation organizations and seed companies.

This article aims to provide an overview of the results and applications of the dry chain approach for both researchers and practitioners interested in accessible climate-smart technologies for medium-term conservation of seeds. To support this objective, this paper has four goals: (1) provide a thorough review of the published literature on the implementation of the dry chain and the use of zeolite drying beads to enhance seed storage; (2) provide additional original unpublished results from experiments on seed conservation that employed zeolite beads in Costa Rica, Mexico, and Nepal; (3) provide results from extension projects on the dry chain in Bangladesh, Pakistan and the South Pacific (Fiji and Tonga); and (4) identify challenges encountered and propose opportunities for successful extension and implementation of the dry chain.

Survey Methodology

The content of this article is divided into three main sections, a literature review of recent published research on the dry chain technology and the presentation of original results on the application and extension of this technology in various countries. In the first section, we reviewed all the currently available and accessible literature (to the best of our knowledge) reporting experimental results utilizing the dry chain technology (defined as: “initial seed drying with a reusable desiccant in the form of zeolite beads followed by seed conservation in hermetic containers”; Guzzon et al., 2020). We did not consider articles dealing only with hermetic storage of seeds or only with the use of desiccants other than zeolite drying beads for seed drying (unless the articles dealt with the comparison of drying beads with other desiccants). We retrieved the articles by searching for “dry chain” or “dry chain technology”, “drying beads” or “zeolite beads” also coupled with “seed” in Google Scholar. Twenty-one papers were identified and the main results of all these publications are summarized and organized, by location, species, protocols used, seed quality parameters tested and main results, in Table S1. With respect to protocols, these primarily refer to the quantities of drying beads used relative to seed weights, containers used for seed storage, and preparation of the desiccants.

The previously unpublished results presented in the second and third section were collected by the authors and are organized by country. The studies in different countries vary in terms of approaches followed and in the scales of application of the dry chain. Some focus more on experimental activities while others focus more on the strategies employed to scale up the application of this technology by farmers and seed handlers. The aim of these sections is to present different case studies of the implementation of the dry chain at different scales, in different areas of the world and involving various groups of stakeholders. We believe that these case studies will be informative for others attempting to implement dry chain methods for seed storage.

Review of Prior Results

As mentioned in the survey methodology, we focused our review on studies dealing with seed drying with zeolite drying beads followed by storage in hermetic containers.

Measure the moisture status of the seeds

First of all, it is important to know the SMC prior to and after drying, but obtaining a precise measurement of SMC is a destructive and relatively slow process by means of a standard oven test, which involves weighing a seed sample, drying in an oven for up to 17 h (depending upon the species), and reweighing to determine the weight loss (ISTA, 2022). Moisture meters can be used for non-destructive measurements of SMC, but these are generally calibrated only for a limited number of crops (Hay et al., 2023). Alternatively, eRH can be measured within a few minutes and non-destructively using a water activity meter or a relative humidity meter (see e.g., Ndinya et al., 2017; MSBP, 2022; Hay et al., 2023). However, such equipment is not readily available in rural communities or to individual farmers. Thus, the use of relative humidity indicator cards is recommended to provide a quick estimate of eRH, which can be readily converted to SMC if needed (Bradford, Dahal & Bello, 2016; Thompson et al., 2017). These humidity indicator strips change color in response to RH and can quickly indicate the RH inside a container containing a seed sample. All that is required to estimate SMC is to insert an indicator strip with the seed sample inside hermetic container (e.g., a clean, dry plastic water bottle), allow it to equilibrate (generally within 30 min), and compare the color to the chart provided with the RH indicator strip/card. For larger quantities of seeds, representative sampling (such as by using a trier to sample large bins) is important (Bradford, Dahal & Bello, 2016; ISTA, 2022). As mentioned, eRH measurements can be converted to MC estimates, and online tools are available to facilitate these conversions for many crop and plant species (see e.g.,  https://ser-sid.org/viability/moisture-equilibrium and http://www.dryingbeads.org/tools).

Introduction to zeolite drying beads

Briefly, drying beads are composed of zeolite crystals that have sites that can tightly bind individual water molecules. Once the sites are occupied, the water can be removed by heating the beads to over 250 °C for approximately 2 h, or until all beads have reached this temperature. The beads are cooled in water-proof containers to prevent reabsorption of water from the atmosphere and then stored hermetically until used. Inside of a closed container, seeds will equilibrate with the RH of the air at the ambient temperature. When beads are present, they will absorb the water from the air, reducing the RH in the container to near zero. The seeds will then lose water to this low RH atmosphere through the vapor phase until the beads reach their capacity or the forces binding water in the seeds become equal to those in the beads. Even an excess quantity of fully activated beads generally will not reduce SMC below about 2–3%, at which point the binding energy of some seed components equals that of the beads. If a specific final MC is desired, the weight of beads to store with a given quantity of seeds at a known MC can be calculated from the amount of water in the seeds and the bead water-holding capacity (Bradford et al., 2018). A drying beads calculator is available to calculate the correct weight of beads to use considering seed quantity, initial SMC (or eRH), ambient temperature and oil content for 69 different crop species in order to achieve a given final MC when the beads are saturated (see: http://www.dryingbeads.org/tools). For larger-scale applications, such as with commercial seeds, a pragmatic approach is to first incubate seeds with 40% of their weight of beads (0.4 ratio). After one or two days, the seed eRH can be easily measured (see also the chapters below on the application of the dry chain in Costa Rica and Nepal); if it is at the desired level, the beads can be removed, and the seeds packaged for storage. If the SMC (or eRH) is still too high, the process can be repeated a second time, which is generally sufficient unless the initial SMC is very high.

Effects of the dry chain on seed moisture content, germination percentage and germination timing

In all the studies reviewed, drying beads quickly lowered the MC of the seeds, without damaging them or affecting germination, except for the results of Trail et al. (2022). In this study, zeolite beads (stored with the seeds) dried okra, sorghum and velvet beans to ultradry conditions (<5% MC), detrimentally affecting seed viability in okra and velvet beans but not sorghum. The authors of this study observed that the ratio of beads to seeds was probably too high, considering the initial low moisture contents of seed samples. This highlights that, when using drying beads as a seed desiccant, it is important to utilize the correct ratio of beads to seeds, based on initial moisture content and specific seed characteristics (e.g., oil content) to prevent over-drying of sensitive species. Moreover, such as for long-term germplasm storage, where lower moisture levels are desirable, the seeds should be incubated at ambient (or higher) RH for several days after removal from storage to hydrate the seeds via the vapor phase prior to imbibition in liquid water. This will prevent imbibitional damage that can occur with some species (e.g., legumes) when they are rapidly imbibed from very low MC (Bewley et al., 2013). This rehydration procedure was not employed in the study by Trail et al. (2022), who noted that the damage observed in germination tests of okra and velvet bean seeds appeared to be associated with imbibitional damage.

Other than the described over-drying risk of sensitive species (Trail et al., 2022), the germination of seeds dried with beads was always higher than the controls (i.e., seeds conserved in open storage and/or undried seeds in hermetic containers) after different periods of storage (Table S1). For example, Bakhtavar et al. (2019) demonstrated that maize seeds dried with beads and conserved in hermetic plastic bags for only 4 months had a faster mean germination time (MGT) when compared with seeds with a higher pre-storage MC and/or conserved in porous containers. A slower MGT is often observed with seed ageing (Demir et al., 2011; Bradford & Bello, 2022). Yahaya, Sinniah & Misran (2022) also showed that seeds dried with beads had faster germination when compared to seeds dried using traditional drying systems or oven drying after six months of storage. Seeds conserved following the dry chain performed better than controls also in terms of seedling vigor (percentage of seedling emergence, seedling length, dry biomass or seedling vigor index) when compared with controls after different storage periods (e.g., Guzzon et al., 2020, six months of storage; Kamran et al., 2020, five months of storage; Yamalle, Khade & Singh, 2020, twelve months of storage; Hilli & Vyakaranahal, 2019, nine months of storage). Plants derived from seeds of two cotton genotypes dried with beads and hermetically stored produced plants with more bolls and sympodial branches, that bloomed earlier, subsequently producing higher yields when compared to sun-dried seed lots (Kamran et al., 2020, five months of storage). Moreover, drying beads enabled faster drying while maintaining high seed viability when compared with traditional methods such as sun drying (Nassari et al., 2014; Kamran et al., 2020; Yahaya, Sinniah & Misran, 2022) or drying with cow dung ash (Sultana et al., 2021).

Effects of the dry chain on physiological indicators of seed ageing

Seeds dried with desiccants and stored hermetically showed reduced seed deterioration, as indicated by physiological indicators of seed ageing. For example, seeds dried with beads and conserved in hermetic containers had lower levels of malondialdehyde (MDA) when compared with undried seeds and seeds stored in porous containers (Bakhtavar et al., 2019, four months of storage; Bakhtavar & Afzal, 2020a, eighteen months of storage; 2020b, four months of storage). MDA is one of the by-products of lipid peroxidation and has been used to quantify seed deterioration due to oxidation (Gianella et al., 2022). Tahir et al. (2023) highlighted that rice seed quality in terms of germination and antioxidant defense mechanisms was preserved when dried to 10% MC with desiccant beads and stored hermetically for six months.

Bakhtavar & Afzal (2020a, eighteen months of storage) found the lowest levels of reducing sugars in quinoa seeds using the dry chain, when compared with poorly dried seeds and seeds stored in porous containers. Reducing sugars, which can interfere with the normal functioning of membranes, are formed during ageing as a result of sugar hydrolysis under high SMC due to the formation of reactive oxygen species (ROS). Therefore, high levels of reducing sugars, together with lipid peroxidation, are among the markers of the biochemical deterioration associated with seed ageing (Murthy & Sun, 2000). Kamran et al. (2020, five months of storage) found that cotton seeds dried with zeolite beads and stored hermetically did not show a significant increment in free fatty acids (FFA). Increases in FFA, due to hydrolyzation of lipids, is another hallmark for seed and grain deterioration (Wang et al., 2020) and was particularly evident in the experiments of Kamran et al. (2020) in seeds stored in porous containers.

Seed quality can also be estimated through conductivity tests that measure electrolytes leaking from seeds. Changes in the organization of cell membranes occur during seed dehydration to preserve them in the dry state (ISTA, 1995). During subsequent imbibition, higher vigor seeds reorganize seed cellular membranes and repair cellular damage quicker and to a greater extent than lower vigor seeds, which corresponds to lower measurement of electrolyte leakage (ISTA, 2022; Marin et al., 2018). Yamalle, Khade & Singh (2020, twelve months), Bakhtavar & Afzal (2020a, eighteen months) and Sultana et al. (2021, nine months of storage) showed that seeds of onion, quinoa and mung beans conserved with the dry chain showed the lowest values of seed leachates upon imbibition when compared with undried seeds and/or seeds conserved in porous containers. High α-amylase activity also indicates vigorous seeds with low deterioration (Marques et al., 2014). Bakhtavar & Afzal (2020a, eighteen months of storage) showed that quinoa seeds stored in hermetic plastic bags exhibited maximum α-amylase activity, thus demonstrating the effectiveness of maintaining low SMC in hermetic bags to preserve seed quality. Moreover, maize and wheat seeds dried with zeolite beads and conserved at low MC in hermetic plastic bags showed the highest starch and protein contents, when compared with control treatments (Bakhtavar et al., 2019, four months of storage; 2020b, four months of storage).

Effects of the dry chain on insect and fungal infestations

Seeds dried to low MC and then stored in hermetic containers also were less susceptible to insect and fungal infestations as well as to mycotoxin contamination (Afzal et al., 2017). Maize seeds dried with zeolite beads and stored in hermetic containers showed reduced insect and fungal infestations when compared to undried seeds in rural communities of the Guatemalan highlands (Guzzon et al., 2020, 6 months of storage). Sultana et al. (2021, nine months of storage) confirmed that the dry chain reduced the frequency of seed borne pathogens in mung beans stored in hermetic containers. Seed conservation with the dry chain approach also greatly reduced infestation by the lesser borer in maize (Bakhtavar et al., 2019, four months of storage), bruchid beetles in mung bean and groundnut (Kunusoth et al., 2012, seventeen months of storage; Sultana et al., 2021, nine months of storage; Singh & Mishra, 2022, six months of storage) as well as rice weevil in wheat (Nelwadker et al., 2022, twelve months of storage). Similarly, the dry chain reduced seed weight loss due to insect damage as well as oviposition and insect respiratory activity in maize and mung bean (Bakhtavar et al., 2019, four months; Sultana et al., 2021, nine months of storage). Singh & Mishra (2022, six months of storage) observed that groundnut pods conserved with zeolite beads reduced weight loss due to insect damage as well as fecundity of bruchids while minimizing losses in seed germination.

The high moisture content of stored grains can also cause increases in mycotoxin occurrence (e.g., fumonisins and aflatoxins) that are responsible for severe health effects in consumers, such as liver necrosis and tumors, depressed immune esophageal cancer, stunting and neural tube defects (Bryła et al., 2013; Wu, Groopman & Pestka, 2014). Bakhtavar et al. (2019, four months of storage; 2020b, four months of storage) determined that drying maize and wheat seeds with zeolite beads and storing them in hermetic plastic bags completely prevented aflatoxin accumulation during storage. Taken as a whole, these results demonstrate that the application of the dry chain can potentially prevent postharvest losses that account for 40% of food losses in developing economies (Claes et al., 2021). In particular, aflatoxin contamination poses a serious health threat to humans and livestock as well as a significant economic burden, causing an estimated annual loss of 25% or more of the world’s food supply (WHO, 2018; Dahal et al., 2023).

Comparison between zeolite drying beads and other desiccants

Some experiments dealt with the comparison between zeolite drying beads and other desiccants, mainly silica gel, which is one of the most used reactivatable desiccants for seeds, due to its low cost and high availability. Hay et al. (2012) demonstrated that zeolite beads showed a higher affinity for water over silica gel, particularly at low MC (<9–10%), being able to dry rice seeds to lower moisture content than silica gel. In fact, silica gel has a higher water-holding capacity than beads at high RH, but its ability to absorb water declines linearly with RH. Thus, at low SMC, where eRH is low, the capacity of silica gel to absorb water is reduced compared to beads. In contrast, drying beads will absorb a specific amount of water across a wide range of RH, and remain absorbent down to very low RH. Thus, their absorptive capacity does not change as the seeds dry, making it easy to calculate the amount of water that they will absorb (http://www.dryingbeads.org/tools). In addition, Kunusoth et al. (2012) highlighted that zeolite beads can be more reusable than silica gel, as they regain their full potential capacity after reactivation, whereas silica gel will gradually lose its capacity with repeated reactivation. Nivethitha et al. (2020) showed that drying beads, at the same bead:seed ratio, can dry okra seeds to lower moisture contents and faster than silica gel. Similar results were obtained by Nassari et al. (2014) in drying tomato seeds with zeolite beads and silica gel. Sultana et al. (2021) confirmed that drying beads, as well as other desiccants such as silica gel, sodium aluminum silicate and activated alumina, can be effective desiccants for mung bean seed storage activities.

Concerns on the use of drying beads

Despite the overall positive results obtained by the zeolite drying beads, some concerns arose from their use. In particular, Hay et al. (2012) and Hay & Timple (2013) reported that beads did not seem to work to full capacity in a bead-seed system, and therefore calculating the quantity of beads to use to reach the target moisture content was not always straightforward. While these observations deserve further research, it is also important to highlight that, if the drying beads are not fully reactivated before use (by heating to >250 °C, for two hours), their absorptive capacity could be less than expected. It is therefore good practice to check the maximum capacity of the beads prior to use if their storage conditions since reactivation are not known. This can easily be done by weighing a small quantity of beads, placing them in a sealed container over water (as on a screen supported above the water), and weighing again after incubating at least several hours, or until weight is constant (Ndinya et al., 2017). The weight gain relative to initial weight indicates the current bead adsorptive capacity, which should be compared to that of recently reactivated beads.

Seed drying protocols and applications

In the publications analyzed (Table S1), different drying treatments were used for various species at a range of initial MCs. Several drying protocols, including different beads-to-seed ratios and storage conditions, were used in the different experiments, with some authors (e.g., Kunusoth et al., 2012; Guzzon et al., 2020) suggesting that to maximize the drying potential of zeolite beads, these should be replaced or reactivated during the drying process. It emerged from our review that, in some experiments, zeolite beads were stored together with seeds (see e.g., Ndinya et al., 2017; Sultana et al., 2021; Nelwadker et al., 2022; Trail et al., 2022), or removed before storage in hermetic containers (Arjun & Pratima, 2014; Guzzon et al., 2020; Bakhtavar et al., 2019). It is not necessary to store the desiccant together with the seeds as long as the seeds, once dried to low MC suitable for conservation, are then packaged in hermetic containers to maintain the MC achieved (Bradford et al., 2018). This means that after the initial drying, beads can be removed, reactivated and reused. This reduces the investment required for the acquisition of sufficient zeolite beads. In addition, it is important to note that bead drying can be combined with other drying systems, particularly air and sun drying. Ambient conditions, whether in sun or shade, are universally used to dry seeds and are effective in removing moisture to equilibrium with the ambient RH. However, as seed eRH cannot be lowered below the ambient RH, this often may not be sufficient to reduce SMC to levels for safe storage. However, the remaining moisture can easily be removed using drying beads to lower SMC to levels safe for sealed storage and extended longevity (see, e.g., the section below on the application of the dry chain in Costa Rica). This reduces the quantities of beads (or reactivation cycles) that would be required to process a given quantity of seeds. An initial stage of air or sun drying is recommended, particularly for seeds that are wet at harvest, such as tomatoes or melons, so that no surface water remains on the seeds if they are to come in direct contact with the beads. When beads absorb liquid water quickly, they emit heat, which can be damaging if beads are mixed directly with wet seeds.

The reusability of the beads is particularly important since it was highlighted that zeolite drying beads are currently logistically and economically inaccessible to individual farmers (Kunusoth et al., 2012; Guzzon et al., 2020; Musebe et al., 2020). It has been proposed that, to increase the access to drying beads, “drying centers” could be organized in farming communities in tropical areas (Dadlani, Mathur & Gupta, 2016; Bradford et al., 2018; Guzzon et al., 2020). These centers should be organized within infrastructures that are already exist and are being used by local farmers, such as community seedbanks or agricultural cooperatives. Drying kits, including beads, hermetic containers, ovens to reactivate the beads and instruments to evaluate MC and/or eRH (such as DryCards, reusable cards with a strip of relative humidity indicator paper embedded in them; Thompson et al., 2017) can be provided to interested institutions to spread the use of this technology. In this context, it would be important to learn from the experience of the “blue drums”, a drying system developed by the Millennium Seed Bank of the Royal Botanic Gardens, Kew (UK), consisting of a sealable, hermetic plastic drum containing silica gel, placed within a central cone made of metallic net from which bags of seeds are hung for drying (Sutcliffe & Adams, 2014). These drums are already being used as a low-tech drying method in seed banks in 47 different countries and territories across the globe (Martens, 2018). In this review of the current available literature, most of the studies on the dry chain were realized in seed laboratories in controlled conditions. More research on strategies for applying this technology in rural communities of low- and middle-income countries in tropical areas, as well as studies on the socio-economic implications of the use of dry chain technology are needed. We provide some examples of different strategies to implement the dry chain in rural communities in the following sections of this paper.

Hermetic containers for seed storage

Several types of hermetic containers were used to store seeds in order to maintain the low moisture content achieved with desiccants, including plastic containers and drums (Afzal et al., 2019; Guzzon et al., 2020; Sultana et al., 2021), plastic bags (e.g., Super Bags, Bakhtavar et al., 2019), laminated aluminum foil packets (Hay et al., 2012; Yahaya, Sinniah & Misran, 2022) and glass jars (Trail et al., 2022). This highlights that diverse types of moisture-proof containers can be used to store dry seeds, and that this approach can utilize a variety of different locally available hermetic containers. While we have concentrated here on seeds, the same conditions apply for grains or other food products. Anokye-Bempah et al. (2023) demonstrated the potential of the dry chain (in terms of storage with beads within hermetic plastic bags) in maintaining the correct MC during storage and long-distance transportation of green coffee beans otherwise exposed to undesirable fluctuations in temperature and relative humidity, without affecting sensorial qualities. This shows the potential of employing the dry chain also for the postharvest storage of other foods and plant materials of great economic importance.

Future applications and research directions

Most of the experiments on the application of the dry chain technology were performed on seeds of food crops (cereals, pseudocereals, vegetables and legumes), due to their direct importance in food security and farmers’ livelihoods. In the future, it would be important to test this technology for the short- and medium-term seed conservation of wild species in local seed banks, restoration seed banks (sensu Merrit & Dixon, 2011) and in reforestation activities in tropical areas, considering that critical hotspots of plant biodiversity are located in tropical areas and that ecological restoration provides a significant opportunity for achieving global conservation goals in the context of climatic change (De Vitis et al., 2020). In this context, a relatively high percentage of plant species in humid regions have recalcitrant seeds that are intolerant of drying, and the species’ desiccation tolerance should be known before dry chain technology is implemented for seed collection, storage or restoration of such species (Wyse & Dickie, 2017).

Examples of Practical Implementation of the Dry Chain Technology Across Tropical Countries

In this section of the paper, we present original, unpublished data on implementation strategies for dry chain technology under different conditions. In particular, we address how to further extend this technology, maximize its impact, make it locally accessible and train relevant stakeholders. We organized these contributions in two subsections: (1) Results of previously unpublished research on the implementation of the dry chain technology conducted in Costa Rica, Mexico and Nepal (these country-specific experimental contributions are organized with an introduction, materials and methods, results and conclusions sections); and (2) examples of extension activities to implement the dry chain approach by presenting brief examples from extension projects to spread the use of this technology in different areas of the world (Bangladesh, Pakistan and the South Pacific).

Research Case Studies on the Implementation of the Dry Chain Technology

Costa rica

Introduction

Costa Rica is characterized by a high RH, especially on the Caribbean side of the country. This consistent high humidity can detrimentally affect the storage of seeds, grains and other dried food products. The Seed and Grain Research Center (CIGRAS, by its Spanish acronym) of the University of Costa Rica (UCR) is conducting research and outreach activities to improve seed and grain storage among farmers. The dry chain technology was introduced to CIGRAS in 2019, including RH indicator cards for measuring eRH (Thompson et al., 2017; Bradford, Dahal & Bello, 2016) and zeolite beads for seed drying (Bradford et al., 2018). Several experiments were conducted to implement the use of the drying beads, test their reactivation and drying dynamics, and assess containers and bags for hermetic storage. The species studied included papaya (Carica papaya L.), maize, rice and beans. In 2021, 250 kg of beads were procured to initiate a series of workshops with communities conserving seeds in Costa Rica. This section describes the implementation and extension of the dry chain in Costa Rica in cooperation with different stakeholders, more in detail: (1) cooperation with the Papaya Seed Production Group to improve the conservation of papaya seeds; (2) cooperation with community seed banks and other local stakeholders to improve the conservation of maize and bean seeds. For this second point, four communities were targeted as potential users to implement the dry chain and share knowledge: (1) “Red Sancarleña de Mujeres Rurales” (RESCAMUR) (number of users, n = 19); (2) the Ujarrás Indigenous group from “Buenos Aires” (n = 10); (3) Nicoya community seed bank (n = 11); and (4) extensionists, academics and researchers (n = 6). The first three stakeholders maintain rural community seed banks. RESCAMUR is a local women’s organization committed to preserve seeds of traditional crops and conserve landraces for self-consumption. The Ujarrás Indigenous group and Nicoya seed bank are associations of small growers, and, along with RESCAMUR, they preserve and exchange seeds of landraces for their own self-consumption and commercialization in the local market. These rural communities are located in regions with high rainfall, temperature and RH (Table 1).

Table 1 Average yearly rainfall, temperature (maximum, minimum), and relative humidity in three locations of Costa Rica where community seed banks are located (period 2012–2022).

Data source: National Institute of Meteorology (IMN, by its acronym in Spanish).

Location	Rainfall (mm)	Temp max (°C)	Temp min (°C)	Relative humidity (%)	
San Carlos	3460.9	30.4	21.4	86.5	
Buenos Aires	3140.7	31.3	20.4	84.9	
Nicoya	1794.5	33.7	23.2	74.4	

Materials and methods

Papaya seeds were received from the plant breeding and seed production program hosted by the UCR and INTA (an agricultural technology transfer center of the Ministry of Agriculture). Seed drying with beads was preceded by air-drying. Seed drying with beads was conducted in hermetic plastic containers for up to 8 days. The beads’ water absorption capacity at 75% RH was ∼20%. Papaya seed moisture content was quantified by placing seeds (∼1 g) in metallic containers and drying them for 24 h at 105 °C. The quantity of beads employed for drying was calculated using this formula, ΔW=MCi−MCf∗initial seed weight in kg/1−MCf

Beads to add (kg)=ΔW/bead capacity as fraction

where ΔW = kg of water to remove, MCi = initial SMC (FW basis) as fraction; MCf = final SMC (FW basis) as fraction; bead capacity is the amount of water as a fraction of their dry weight that the beads can absorb (∼0.2). Seed germination tests were conducted in an incubator at 30 °C, using peat moss in plastic containers at nearly 100% RH, with 12 h dark/light cycles. Before testing, seeds were allowed to equilibrate their SMC with the environment humidity by opening the plastic bags in which they were stored for at least two days. Seeds were then soaked in gibberellic acid at a concentration of 500 ppm for three hours. Counting of normal seedlings was started 10 days after sowing and finished on day 21 after sowing.

As mentioned, efforts were also made to implement the dry chain in Costa Rica by working with community seed banks and other local stakeholders on drying and conservation of maize and bean seeds. In particular, the use of RH indicator stripes was tested as well as seed drying with zeolite beads. The germination tests with maize and bean seeds in the communities were carried out in transparent plastic boxes with two replicates of 50 seeds sown in an organic substrate (peat moss). Germination was scored at 7 and 8 days after sowing for maize and bean, respectively.

Results

The Papaya Seed Production Group was among the first users of the dry chain technology in Costa Rica. Papaya seeds are produced in two regions of Costa Rica, one located in the Caribbean side of the country and the other in the Central Valley. The MC of the seeds available varied consistently between the two locations, with an average of 12.4% (±0.1 SD) for the Caribbean location and 9.0% (±0.03) for the Central Valley location. A second independent group of seed lots was tested, and the difference was even larger, ranging from 14.7% to 9.3%. This clearly indicated the necessity, especially for the location on the Caribbean side, to include a drying process in the seed production system. Ellis, Hong & Roberts (1991) reported that for papaya seeds, storage at a moisture content between 7.9% and 9.4% and 15 °C should maintain original seed germination capacity for a year. Following these results, a target of 8.5% seed moisture content was set for the papaya seeds. The Cromarty, Ellis & Roberts (1982) equation (with a linear R2 = 0.96 with the SMC; Fig. 1A) was used to obtain the eRH at which the SMC would be ∼8.5% (using 30% seed oil content for ‘Pococí’ hybrid). In preliminary experiments it was observed that drying the papaya seeds to 3.02% did not affect the germination (Fig. 1B). Once dried to the desired eRH (∼50%), the seeds were packed in hermetic plastic bags and the seed germination was checked before and after drying with no changes (Fig. 1C).

Figure 1 Dry chain experiments in Costa Rica (1).

(A) Relation between seed moisture content (SMC) on a dry weight (dw) basis and estimated SMC using the Cromarty equation in Carica papaya. (B) Relation between SMC and seed germination after drying using zeolite beads. (C) Germination before (purple bars) and after drying (blue bars) with zeolite beads in different seed lots (A–D).

As previously mentioned, in the three community seed banks involved in this project (see Fig. 2), the most commonly grown crops are maize and beans. One key need was to provide the communities with an affordable method to know whether their seeds were dry enough for storage. For this, an inexpensive dry-circle RH indicator card was designed and produced with instructions in local language (Figs. 2B and 2C). The main intentions in the design of the dry-circle card were to avoid using plastic and to maximize the reuse of the indicator to limit costs. At first, in all communities, all the dry-circles incubated with seeds exhibited pink color (Figs. 2B and 2D), indicating high eRH of the seeds, consistent with the high ambient RH in these locations (Table 1). Thus, seed drying beyond open-air drying was needed, so the communities were instructed on how to store zeolite beads with the seeds in a sealed container (beads equal to 40% of the total weight of the seeds, 0.4 ratio) for two days before storing the seeds in a hermetic container. An average of 9% and 11% SMC for maize and beans was detected after drying these samples. The training also included evaluating seed physiological quality employing a seed germination test. The germination tests were carried out two weeks after the storage in the hermetic containers, showing 91% germination in maize (average of five samples) and 78% in beans (average of 27 samples). However, in beans the values ranged from 62% to 94%, indicating a range of initial viabilities among the accessions.

Figure 2 Dry chain implementation in Costa Rica.

(A) Locations of the seed communities in Costa Rica using dry chain tools (the map was generated with ggplot 2, version 3.4.2, using Rstudio 2023.12.0 “Ocean Storm”). (B, C) Wet (B) and dry (C) rice samples, both containing a dry-circle indicator inside (photos: Ester Vargas). (D, E) Bean seed accessions from the community of Nicoya (photos: Andrés Monge). (F) Seed sanctuary from Rescamur community (photo: Andrés Monge).

Conclusions

The stakeholders involved in this project have found the implementation of dry chain technology to be important and valuable. The studies conducted to demonstrate that drying using beads had no detrimental effects on initial seed quality were useful in promoting adoption of the method. Similarly, the use of the dry circles in the community seed banks to easily indicate eRH was immediately appreciated by the participants. The activities conducted since 2019 have given them support to organize and improve the quality of community seed banks (Figs. 2E–2H). One of the main limiting factors for the full implementation of the dry chain remains the local unavailability of zeolite beads. Reactivation of the beads by heating has been conducted by CIGRAS. Moisture-saturated, inactive beads are received and exchanged for dry, active ones. However, access to relatively inexpensive electric or gas ovens could enable local communities to reactivate beads themselves. In the future, improving local supplies of zeolite beads and follow-up trainings will be necessary. In the work with extensionists and academics, the use of the dry chain has been important to decrease moisture contents of orthodox seeds and preserve germplasm. At CIGRAS, long-term seed storage experiments with beans, maize, and rice are being conducted, following a similar experimental design as the one conducted by Guzzon et al. (2020).

Mexico

Introduction

The dry chain technology was introduced at the International Maize and Wheat Improvement Centre (CIMMYT, Texcoco, Mexico) in 2015 to support community seed banks in Guatemala and Mexico (see e.g., Guzzon et al., 2020). Moreover, the application of the dry chain was also tested within the CIMMYT Germplasm Bank itself (the largest germplasm collection of maize and wheat genetic resources in the world) as a potential opportunity to enhance seed processing and drying in the tropical research stations in Mexico where the regeneration of tropical maize material is carried out, especially in the research station of Agua Fría (Venustiano Carranza, Puebla State; hereafter referred as to AF) where a high occurrence of fungal and insect infestations as well as rapid losses of seed viability were observed. This research station has an old heat-based seed drier (6 DOOR; STONE CONVEYOR Co. Inc., Honeoye, NY, USA, 1967) that was no longer in use. In 2018 the heating system was removed and the drying cabinet underwent some renovations (painting and porous trays to allow air flow were added; see Fig. 3A). The aim of this project was to test the use of drying beads coupled with the air flow generated by the ventilator of the cabinet to enable fast drying of the seeds before they were transported to CIMMYT Genebank for the last phases of seed processing and packing (El Batán, Texcoco, Mexico State, hereafter referred to as GB), without adversely affecting seed longevity in storage.

Figure 3 The cabinet used for seed drying in CIMMYT’s Agua Fria Research Station (Mexico).

(A) The drying cabinet in 2019, after renovation work, with porous trays in the middle of the sections where mesh bags with seeds are placed, as well as metal grids at the base of each section where mesh bags with beads are placed (photos: DE Costich). (B, C) Schematic views of the drying cabinet.

Materials and methods

In order to compare the two drying methodologies, the same quantities of seeds of 21 maize accessions (see Table 2 for the initial weight and MC of the accessions employed in this research) were dried in AF with the use of beads in the drying cabinet as well as in GB in a conventional dry room. Additionally, the seeds of the same accessions and origin dried in AF and GB with the two methods to similar MC (see Table 2 for MC values after drying) were then exposed to accelerated ageing to assess whether the drying with beads had influenced potential seed longevity. A detail description of the methodology employed for this experiment is presented in Methods S1.

Table 2 Maize accessions tested in drying experiments at CIMMYT.

The CIMMYT Accession ID, the initial weight of each accession, the moisture content (MC) of each accession before the drying treatments, the final MCs of the two seed lots per accession exposed to the drying treatments as well as the days taken by the seed lots dried in the conventional dry room at GB to reach the target MCs are indicated. The p50s of the seeds dried with the two methods are also presented for the 12 accessions exposed at accelerated ageing. The asterisk near the accessions ID indicates the accessions for which the final MC after drying at GB was outside the 95% confidence range of the MC after drying with beads in AF.

Accession ID	Initial Weight (grams)	Initial MC (%)	MC after drying AF (%)	MC after drying GB (%)	Days of drying GB	p50 AF (days) ± s.e.	p50 GB (days) ± s.e.	
CIMMYTMA 31977*	2,248	16.15	8.74	8.5	15	15.95 ±1.07	15.91 ±1.08	
CIMMYTMA 31979	2,156	10.39	8.46	8.41	11	12.08 ±1.09	11.6 ±1.09	
CIMMYTMA 31995*	4,214	15.33	8.6	8.44	21	17.08 ±1.07	15.9 ±1.07	
CIMMYTMA 32008*	2,556	13.06	8.57	8.17	15	12.56 ±1.09	14.73 ±1.08	
CIMMYTMA 32012	2,576	17.63	8.2	8.21	15	20.38 ±1.06	20.27 ±1.07	
CIMMYTMA 32019	2,123	17.02	8.5	8.58	15	17.11 ±1.08	17.14 ±1.08	
CIMMYTMA 32037	3,810	16.02	8.6	8.71	15	18.71 ±1.07	19 ±1.06	
CIMMYTMA 32040	3,134	15.08	8.62	8.64	17	15.06 ±1.07	19.86 ±7.78	
CIMMYTMA 32044	4,000	10.81	8.37	8.26	15	11.89 ±1.09	14.7 ±1.07	
CIMMYTMA 32064	3,160	9.90	8.3	8.28	11	22.51 ±1.06	19.2 ±1.07	
CIMMYTMA 32070*	2,858	10.45	8.72	8.41	11	21.52 ±1.06	20.25 ±1.07	
CIMMYTMA 32120	3,786	15.43	8.79	8.89	15	20.21 ±1.07	25.24 ±1.05	
CIMMYTMA 32122	4,035	13.15	8.66	8.65	17	–	–	
CIMMYTMA 32123	3,018	10.46	8.43	8.47	11	–	–	
CIMMYTMA 32126	4,310	11.09	8.43	8.43	17	–	–	
CIMMYTMA 32136	3,376	11.85	8.6	8.55	15	–	–	
CIMMYTMA 32179	2,700	10.02	8.12	7.97	15	–	–	
CIMMYTMA 31990	4,266	10.16	8.08	8.23	15	–	–	
CIMMYTMA 32033	3,966	11.64	7.87	8.00	17	–	–	
CIMMYTMA 32055	3,600	11.67	8.16	8.32	15	–	–	
CIMMYTMA 32088	3,694	11.73	7.92	7.76	15	–	–	

The 21 accessions employed for this experiment were regenerated at the high-altitude CIMMYT research station in Metepec (Toluca, Mexico State) in 2019. These accessions were sown on 14 March 2019, and the harvest occurred between 10 October and 26 November 2019. The harvested cobs were then brought to CIMMYT GB, air dried and shelled, and stored under room conditions in porous containers until the beginning of the experimental phase in January 2020. Just before the start of the comparative drying trials on 20 January 2020, the MC of each accession was tested with three replicates per seed lot using a moisture meter and the seeds of all accessions were weighed (see Table 2) and then divided into two seed lots of the same weight. One seed lot per accession was put in the dry room of GB at 9–15 °C and 10–20% RH. The other seed lots were brought to AF to be dried with the beads in the drying cabinet. Drying beads were put in mesh bags, which were placed on the metal grid at the bottom of each section of the drying cabinet (see Fig. 3B), for a total of 32 kg of beads. The seed lots were organized in the upper porous trays in the cabinet (see Fig. 3). The mesh bags of drying beads in the cabinet were replaced daily with bags of beads that had been reactivated in the oven of the kitchen of the research station at 200–250 °C for 2–3 h.

At the end of four days of drying in AF, the seeds were packed in heat-sealed trilaminate aluminum pouches. The average temperature and RH in the drying cabinet at AF (measured with dataloggers) were 27.5 ± 2 °C and 21.4 ± 5% RH, respectively. The seed lots in the dry room at GB were dried until they reached a MC value within the 95% confidence interval of the MC of the seed lot of the same accessions dried at AF. After reaching the appropriate MC in the dry room, these seed lots were also packed in trilaminate aluminum pouches.

The accelerated ageing experiment was carried out on 12 accessions randomly selected (see Table 2) and started on 31 July 2020 in the Seed Laboratory of the CIMMYT Germplasm Bank (GB). A total of 420 seeds of each of the seeds lots (dried at AF and GB) were placed in open petri dishes randomly interspersed in two 300 × 300 × 130 mm sealed electrical enclosure boxes (Ensto UK Ltd, Southampton, UK) placed in a compact incubator (UN160; Memmert, Schwabach, Germany) at 60% RH and 45 °C in the dark. The target 60% RH was controlled by placing the Petri dishes over a LiCl solution (see Hay et al., 2008; Newton et al., 2009).

Germination of each seed lot was tested with duplicates of 30 seeds each at the beginning of the experiment and with seeds retrieved from the boxes after 9, 21, 28, 37, 44 days of ageing. See Guzzon et al. (2021) for the description of the germination protocol employed at the CIMMYT maize collection. Germination scoring was performed 1 week after sowing , according to ISTA (2018) criteria.

Statistical analyses and data visualization were carried out in R version 4.2.3 and RStudio 2023.06.0. Binomial GLMs were used to extract the p50s (the time for viability to decline to 50%) using the probit link function. The p50s for each accession and drying method are presented in Table 2. A binomial GLM with probit link function was also applied to determine the effect of drying method, accession and their interaction on the p50 as longevity correlate. A post-hoc multiple comparison analysis (Tukey) was applied to evaluate the differences between drying method for each of the accessions analyzed.

Results

The final MC values of the seeds dried at GB as well as the time (in days) taken to reach the target MC in the conventional dry room (this ranged from 11 to 21 days) are shown in Table 2. The different accessions had significantly different p50s (p < 0.001, Deviance: 213.84, Df.: 11) while the effect on longevity estimates of the different drying methods as well as the interaction of drying method and accessions were not significant (p = 0.63, Deviance: 0.22, Df.: 1 & p = 0.4, Deviance: 11.48, Df.: 11). The longevity estimates from the two drying methods were not significantly different in any of the accessions (p > 0.05). The raw data of the accelerated ageing experiment are presented in Table S2.

Conclusions

The results of this experiment highlighted that the use of activated drying beads coupled with air flow can quickly dry large seed lots without any detrimental effects on seed longevity, estimated through accelerated ageing, when compared to equilibrium drying in a conventional dry room following international standards for long-term seed conservation (FAO, 2014). Further research with more accessions and crop species as well as different methods to estimate seed longevity can be performed to ensure that this drying methodology can be integrated into the processes of a genebank without affecting the long-term conservation of stored accessions.

Nepal

Introduction

Several experiments on the use of dry chain technology in different crop species were performed in Nepal by both private and public institutions starting in 2012. Some of these experiments were conducted in collaboration with iDE, an international nonprofit organization, the non-profit Center for Environmental and Agricultural Policy Research, Extension and Development (CEAPRED) and Nepal Agriculture Research Council (NARC) (Arjun & Pratima, 2014).

Materials and methods

The dry chain technology was evaluated on several crops, including onion, okra, bean and cucumber seeds (Figs. 4A, 4B, 4C and 4D, respectively). The experiments included comparison between seeds stored in cloth bags and in plastic hermetic containers with drying beads (beads equal to approximately 40% of the total weight of seeds, or 0.4 ratio) for up to 20 months under ambient temperature and RH conditions. Germination was tested at the Seed Quality Control Center of Nepal using standard paper towels seed testing methods at 25 °C.

Figure 4 Seed germination and viability during storage in different areas and communities in Nepal.

Experiments were managed by CEAPRED on onion (A), okra (B), bean (C) and cucumber (D) seed. Most experiments were conducted in mid-hills and cooler regions within Nepal (circles and continuous lines) but also in plains and warmer regions (triangles and dotted lines).

Results

Storage with drying beads lowered eRH of onion seeds from 70.6% (±11.3 SD) to 20% or lower (minimum reading for measuring instrument) and hermetic storage maintained low seed eRH at two sites near Bhairahawa (low elevation), two sites in Tansen (mid-hills), Kavre, Kathmandu and Rukum (mid-hills) districts. Control seeds in porous bags showed variable seed eRH as ambient RH varied with the seasons. At all tested locations, initial germination averaged 84% (±6.36 SD) and storage with beads in hermetic plastic containers maintained viability levels between 66%–85% (75% ±6.82 SD) up to 20 months, while the final germination averaged only 10.9% (±13.3 SD) in the control treatment (Fig. 4A). The decline in seed viability in the controls was more rapid in warmer sites in the plains than in the relatively cooler sites in the mid-hills (Fig. 4A, dashed lines with triangle shapes), similar to the results of Guzzon et al. (2020) in Guatemala.

Okra seed experiments conducted at three sites in Kavre in Nepal (Dhulikhel, Methinkot and Amaltari) displayed the benefit of the drying beads on seed quality during storage by reducing eRH from 78% (±5.2 SD) to 23.3% (±2.08 SD) and maintaining seed viability at 76.9% (±2.76 SD) as compared to 33% at Dhulikhel, 11% at Methinkot and 46% at Amaltari in the control treatments (mean of 30% ±17.7 SD) (Fig. 4B).

Bean seed eRH was reduced from 78% (±5.2 SD) to 27.7% (±2.52 SD) with beads at the different sites as compared with weather-dependent variable eRH in control treatments, measured at 62.3% (±3.06) at 18 months. Germination was maintained at 90.6% (±3.36 SD) with beads while control seeds in cloth bags exhibited declining viability ranging from 78% (±1.41 SD) at Dhulikhel and Amaltari to 21% at Methinkot at 18 months of storage (only 1 replication tested due to the lack of control seeds) (Fig. 4C). Beads also lowered eRH of cucumber seeds at the same sites from 78% (±5.20 SD) to 22.7% (±2.08 SD). Some containers with beads leaked initially, so the seeds were transferred to better hermetic plastic containers and the beads were replaced. Germination remained high after 18 months of storage at 95.9% (±2.15 SD) with the drying beads, compared to 75% (±11.7 SD) in the controls (Fig. 4D). The relatively high germination in the control treatment in cucumber could be attributed to conservation in the cooler mid-hills of the Himalayas where the experiments were performed.

Conclusions

These trials in Nepal confirmed the benefits of the dry chain methodology. All involved stakeholders, including farmers and officials, were fully convinced of the importance of drying and waterproof storage containers. Many district-level orientation trainings were held on drying bead technology and the dry chain approach. Several of the experiment participants managed storage of cereals and vegetable seeds in very challenging regions for maintaining seed viability in open storage. The successful results of activities during training sessions, as well as long-term experimental outcomes described here, activated a committed community that could benefit directly from and further disseminate the dry chain approach in Nepal.

Examples of Extension Activities to Implement the Dry Chain Approach

Bangladesh

With support from the Horticulture Innovation Laboratory of the United States Agency for International Development (USAID), Rhino Research (Bangkok, Thailand) conducted projects aiming to create the foundation for spontaneous diffusion and large-scale adoption of drying technologies in Bangladeshi agriculture. The lack of efficient drying and storage systems in the hot, humid climate of Bangladesh poses a significant challenge to seed production and dissemination (Alam et al., 2018). Bangladeshi seed companies estimated that they lost 5–10% or more of their seeds from rapid ageing due to insufficient drying, a percentage that is worth tens of millions of dollars in horticultural seeds alone.

In contrast with many agricultural development projects that focus first at the individual farmer level, the concept of this project was to focus on the strongest links in the seed supply chain: the seed-producing companies. The approach was to encourage the major Bangladeshi seed production and agricultural processing companies to adopt this technology, enabling it to diffuse through commercial channels throughout the sector, including to smallholder farmers.

The project was initiated by signing Memoranda of Understanding (MOUs) with various Bangladeshi organizations, including Lal Teer Seed Ltd., Getco Agro Vision, Metal Seed Ltd. and Development Alternatives Incorporated (DAI). These companies were offered complete hands-on training of their key quality control employees on the use of zeolite bead-based seed drying methods. Seven week-long trainings enabled the trainees to implement these methods in their own facilities and follow up with questions and further training in the subsequent sessions. This enabled each company to adapt the basic technology to their own crop species, processes, and clients. A key adaptation by Lal Teer, for example, was to provide their seed growers with activated drying beads and hermetic containers, rather than require the growers to purchase them. The growers were instructed to place the seeds into the containers with the beads provided and send the sealed containers back to the company. The seeds then arrived at the company already dried and ready for further processing and storage, and the beads could be efficiently reactivated for reuse in company facilities. This system proved to be highly efficient, particularly in the high humidity conditions prevailing in Bangladesh that make air drying relatively ineffective. As a result, these companies have completely adopted the drying beads technology with good results.

Moreover, the government of Bangladesh has proactively adopted the concept of dry chain technology aimed at providing the necessary training, equipment and tools to the local seed producing communities. In support of this, the Bangladeshi Ministry of Agriculture (BAM) directed its public research organizations, such as Bangladesh Rice Research Institute (BRRI), Bangladesh Agriculture and Development Cooperation (BADC), and Department of Agricultural Extension (DAE), as well as academic institutions, to dry and store their high value seed using the dry chain. The Bangladeshi government has initiated a program to provide these basic supplies to rural communities and to individual farmers. They have successfully trained more than five hundred farmers through a funded project and equipped them with necessary dry chain tools to earn better livelihoods with improved seed quality. Apart from the government organizations, the Bangladesh Seed Association (BSA), representing the private seed sector, has emerged as an active support organization for all of the seed companies and seed-producing communities. BSA organizes various training workshops focusing on dry chain technology and continuously emphasizes its adoption through seminars, workshops, or webinars. Keeping in view the keen interest of all the key stakeholders of Bangladeshi seed sector, a new company named Rhino Bangladesh has recently been established by Rhino Research in collaboration with Lal Teer. The newly developed company has already started commercial production of Dry Chain commodities and is successfully meeting the needs of local and international seed market.

Pakistan

As previously mentioned above in the literature review section and in Table S1, researchers from University of Agriculture, Faisalabad (UAF) have widely tested the use of zeolite beads for seed drying and seed storage in diverse hermetic containers across different regions of Pakistan and with numerous crops. In addition to these research activities, UAF have also promoted the use of the dry chain technology among farmers at four locations (Athmuqam, Hattian Bala, Muzaffarabad and Rawlakot) of Azad Jammu and Kashmir, Pakistan. In these locations, UAF researchers demonstrated that germination and integrity of maize seed were maintained better in hermetic bags as compared to conventional bags at all locations (Khalid et al., 2024). The incidence of mold and insect infestations and aflatoxin contamination was also monitored in the stored seed. Maize seed preserved at low seed moisture contents (<12%) in Purdue Improved Crop Storage (PICS) bags (moisture- and oxygen-proof) showed higher seed quality, maintaining dryness during storage (Afzal et al., 2017). Educational and technical workshops were organized to share results and learnings from dry chain experiments with public and private sector stakeholders, including farmers.

Additionally, the dry chain technology was tested and adopted by vegetable seed companies in Pakistan that are currently utilizing desiccant beads for preservation of high value vegetable seeds. In particular, a study (also at UAF) on the use of zeolite drying beads was carried out on onion seeds, which is relatively short-lived in storage when compared to other crops (Walters, Wheeler & Grothenius, 2005). Onion seeds (100g for each experimental unit, 8% was the initial SMC) were stored in hermetic plastic containers after equilibrating at 6, 8, 10 and 12% MC. The MC of 6% was achieved by drying the seeds with desiccant beads, calculated by using the abovementioned drying beads calculator tool. The MCs of 10 and 12% were achieved by adding a calculated amount of water to the seeds, mixed thoroughly in an airtight container. The following equation was used to calculate the amount of water required to increase the seed moisture content of crop seeds up to desired moisture levels (initial and final SMC values are in percentage and on a fresh weight basis and initial seed weight is in grams).

Amount of water required (ml) = [((100 − Initial SMC)/(100 − Final SMC)) × Initial seed weight] − Initial seed weight.

In this experiment it was found that onion seed with 6% MC conserved in hermetic plastic container had generally superior seed quality when compared with seed stored in conventional cloth bags. Seeds stored in cloth bags at 10 and 12% MC showed less than 20% germination (tested in Petri dishes with filter paper at in an incubator at 25 °C for 10 days). Seed moisture content was maintained in hermetic plastic containers while MC fluctuated in cloth bags at ambient RH after six months storage (Fig. 5).

Figure 5 Results of the drying and storage experiment on onion seeds in Pakistan.

Germination (A) and moisture contents (B) of onion seeds having different seed moisture contents stored in hermetic and cloth bags for six months.

The researchers from UAF also conducted several additional experiments to identify the most suitable hermetic containers for Pakistani farmers and seed producers. Several studies were conducted on seeds of vegetable crops, cereals, moringa (Moringa oleifera Lam.), oilseeds and pulses, testing multiple seed traits to evaluate seed quality (e.g., accelerated ageing, electrical conductivity of seed leachates, germination, insect infestation, grain damage and weight loss, MDA content, SMC, oil and protein contents, total soluble sugars, mycotoxin contamination). These studies clearly revealed that porous woven polypropylene and cloth bags are not recommended for seed and grain storage in the season of high RH and that the dry chain through hermetic storage of seeds at low moisture contents in different types of containers (PICS and Super bags, plastic drums and glass jars) helps to preserve seed quality throughout the supply chain (see e.g., Afzal et al., 2019; Afzal et al., 2020; Khalid et al., 2024).

To enhance the conservation of cereal seeds after harvest in spring season when weather is relatively dry (RH < 60%), hermetic bags were introduced to the farmers of South Punjab and Sindh with the support of NGOs (non-governmental organizations). This helps to protect stored grains from insects and fungi, and therefore mycotoxins, during the monsoon season when RH exceeds 90%. In the desert areas of Sindh, smallholder farmers are facing postharvest losses of cereal grains and pulses due to coastal rains that raise RH during the hot summer season; seed quality rapidly deteriorates in these extreme environmental conditions. In these areas, capacity-building initiatives were conducted to equip farmers with the knowledge and skills necessary for the proper implementation and maintenance of the dry chain technology. UAF collaborated with national and international NGOs, district governments, and farmer cooperatives for further technology dissemination in areas prone to and affected by natural disasters. In recent floods (2022), about 33 million people were affected and farmers had no wheat seed for the next growing season. The concept of dry chain technology was introduced to resource-poor farmers of flood-prone areas. The farmers received training on strategies to strengthen their own seed production as well as storage through use of hermetic systems. Low-cost hermetic drums of high-density polyethylene with a built-in digital hygrometer in the lid to measure eRH of stored seeds were introduced to the farmers and seed industry (Afzal et al., 2019). Farmers in flood-affected areas gained access to hermetic containers in order to safeguard their seeds from floods, excess moisture, pests, and diseases. As a result, the loss of viable seeds was reduced, ensuring a reliable seed supply for future planting seasons. These results highlight how the dry chain technology can have a significant positive impact on the livelihoods of smallholder farmers of humid regions of Pakistan.

South Pacific: Fiji and Tonga

Dry chain technology is being tested and used in the Pacific Island Countries and Territories (PICTs) since June 2022 as part of the work of the Seed Laboratory of CePaCT (Centre for Pacific Crops and Trees, Suva, Fiji), the largest genebank of the Pacific region, belonging to the Pacific Community (SPC). Drying beads, RH indicator cards, dataloggers and hermetic plastic boxes were acquired and are now being tested for drying of orthodox tree seeds at CePaCT as part of initiating long-term seed conservation activities at the Centre. After drying, seed lots are packed in heat-sealed aluminum pouches and conserved in freezers at −18 °C. Several experiments are ongoing at CePaCT, in collaboration with the Fiji National University (FNU), in order to: (1) test dry chain technology for the long-term storage of native and endemic Pacific tree seed species, as well as exotic species of economic importance; (2) compare seed drying with silica gel and drying beads; and (3) promote the use of this technology for small-scale seed drying for genebanks, seed enterprises and community seed banks in the PICTs.

As part of this activity, a one-week training event on seed conservation (12 participants) was carried out by CePaCT staff at MORDI Tonga Trust (Mainstreaming of Rural Development Innovation, Nuku’alofa, Tonga) in April 2023 and the dry chain was introduced to the Kingdom of Tonga (including drying beads, RH indicator cards and dataloggers). MORDI Tonga Trust currently conserves 94 accessions of 21 crops (including local Tongan varieties as well as commercial cultivars) in aluminum pouches and hermetic plastic buckets in its seed vault at 5 °C. These accessions are distributed to local farmers and growers across the Kingdom either as seeds or as plantlets grown in MORDI’s nurseries. The dry chain is being tested to enhance the drying of the seed accessions prior to long-term conservation in the seed vault, which at the moment is the only seed cold storage facility in the Kingdom.

Challenges and Opportunities

The review of published reports and previously unpublished data provided in this paper indicate that dry chain technology, including the use of zeolite drying beads and waterproof storage containers, significantly extends high seed quality and viability during storage of many crop species in different tropical areas. This technology can therefore significantly enhance seed security for farmers in many tropical countries. Here we describe some challenges and opportunities that could guide further applications of this technology.

• Zeolite drying beads, as well as the means of reactivating them, are still inaccessible in many areas. Supply chains and local dealers have not been established in most locations where they would be most useful. Provision of these resources by seed companies or the government directly to farmers, as has been done in Bangladesh, is an effective way to reach small farmers who may not have the means to invest in this technology individually.

• In this scenario, a network approach where used drying beads can be exchanged for reactivated ones at central community, public service or company nodes (Bradford et al., 2018; see also the sections on Costa Rica and Bangladesh) can be an efficient use of resources, as farmers only require drying beads at harvest time, allowing a fixed quantity of drying beads to serve larger numbers of farmers of different crops maturing at different times. Stronger collaborations among all the different actors involved (e.g., seed companies, central/local governments, NGOs, genebanks) in promoting this approach could be an important strategy for spreading this technology and its beneficial impacts on local seed systems. After initial introduction and adoption of dry chain methods, local agro-services dealers would recognize the opportunities and provide ongoing supply needs.

• The use of zeolite drying beads as desiccant for seed drying proved very useful for drying of relatively small (e.g., <50 kg) seed quantities at the same time in one hermetic container. As drying is relatively rapid (a few days), larger quantities of seeds can be processed through this system as the drying beads can be reactivated for reuse. This batch-based process has sufficient capacity for handling most horticultural seeds. For much larger seed quantities, such as cereal grains, sun or heated-air drying is still required, although the use of waterproof storage containers after drying using hermetic bags is still essential. Further research is needed to engineer the use of drying beads for drying large seed quantities (see the section on Mexico, coupling drying beads with air flow). Designs have been developed for continuous flow systems that could simultaneously utilize drying beads and reactivate them, enabling drying of much larger quantities of seeds or grains. An opportunity for humid regions would be to invest in practical development of such systems rather than purchasing heated-air driers that perform poorly at high temperature and humidity.

• Despite numerous scientific papers published on the dry chain, there is still some uncertainty on the best protocol to employ to maximize the benefits of the dry chain and related tools. We have tried to fill this gap in this paper by proposing solutions that have succeeded in practice. In particular, international organizations and governmental institutions involved in supporting farmers and agricultural systems in humid regions should adopt the dry chain as a valuable and sustainable practice and include it in their outreach subsidy programs. Such organizations could also take the lead in creating accessible protocols and training materials based on experience to date to better guide the final users.

• In addition to the drying of crop seeds, the dry chain and drying with zeolite beads have also shown important potential for use in other activities from the conservation of native orthodox plant species (see e.g., the section on Fiji) to the storage of grains and other food products in tropical areas (e.g., spices, dried fruits, etc.). Further development of practical applications as well as research on additional plant species and locations are needed to fully exploit the potential of this technology.

Conclusions

The dry chain is a very versatile approach for seed and food storage that can be adapted to local conditions and applied at very different scales, i.e., seed companies, farmers’ cooperatives, community seed banks and genebanks at regional, national and international levels. As reviewed and presented here, the effectiveness of the dry chain has been confirmed through research conducted around the globe. It is time to focus on distributing and implementing this technology in the regions and for the farmers and consumers who are in most need of it. Governments, NGOs and the seed and food industries all have important parts to play in this process.

Supplemental Information

Table S1 Synthetic review of scientific publications on the use of the dry chain approach and zeolite beads for seed drying and conservation

Articles are listed in chronological order.

Table S2 Raw data of the accelerated ageing experiment

Supplemental Information 3 Detailed materials and methods of the Mexico chapter

Detail methodology of the drying and accelerated ageing experiments performed at CIMMYT (Mexico).

We are grateful to the CIMMYT Genebank Team in Agua Fria for their help in carrying out the drying experiment in Mexico. We also would like to thank Fiona Hay (Aarhus University) for providing the enclosure boxes used for the accelerated ageing tests at CIMMYT. The authors of the Bangladesh chapter sincerely acknowledge the technical assistance of the Horticulture Innovation Lab (University of California, Davis, USA) for the Bangladesh project. The authors are also grateful to Costa Rican community seed banks for their work and interest in the dry chain project, and also thank Marcelo Murillo who provided technical assistance in the experiments. We also acknowledge the collaboration of CEAPRED, iDE, Nepal Agriculture Research Council, FORWARD, and CIMMYT in Nepal.

Additional Information and Declarations

Competing Interests

Author Contributions

Data Availability

Johan Van Asbrouck is the president of Rung Rueng Consulting (Rhino Research, Thailand), a technology provider that sells items that are employed for dry chain seed conservation activities. All the other authors have no competing interests.

Filippo Guzzon conceived and designed the experiments, performed the experiments, analyzed the data, prepared figures and/or tables, authored or reviewed drafts of the article, and approved the final draft.

Denise E. Costich conceived and designed the experiments, authored or reviewed drafts of the article, and approved the final draft.

Irfan Afzal performed the experiments, authored or reviewed drafts of the article, and approved the final draft.

Luis Barboza Barquero performed the experiments, prepared figures and/or tables, authored or reviewed drafts of the article, and approved the final draft.

Andrés Antonio Monge Vargas performed the experiments, prepared figures and/or tables, authored or reviewed drafts of the article, and approved the final draft.

Ester Vargas Ramírez performed the experiments, prepared figures and/or tables, authored or reviewed drafts of the article, and approved the final draft.

Pedro Bello performed the experiments, analyzed the data, prepared figures and/or tables, authored or reviewed drafts of the article, and approved the final draft.

Peetambar Dahal performed the experiments, authored or reviewed drafts of the article, and approved the final draft.

César Sánchez Cano performed the experiments, authored or reviewed drafts of the article, and approved the final draft.

Cristian Zavala Espinosa performed the experiments, authored or reviewed drafts of the article, and approved the final draft.

Shakeel Imran performed the experiments, authored or reviewed drafts of the article, and approved the final draft.

Soane Patolo performed the experiments, authored or reviewed drafts of the article, and approved the final draft.

Tevita Ngaloafe Tukia performed the experiments, authored or reviewed drafts of the article, and approved the final draft.

Johan Van Asbrouck performed the experiments, authored or reviewed drafts of the article, and approved the final draft.

Elina Nabubuniyaka-Young performed the experiments, authored or reviewed drafts of the article, and approved the final draft.

Maraeva Gianella performed the experiments, analyzed the data, authored or reviewed drafts of the article, and approved the final draft.

Kent J. Bradford conceived and designed the experiments, authored or reviewed drafts of the article, and approved the final draft.

The following information was supplied regarding data availability:

The raw data is available in the Supplemental Files.

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
