# Peer review of "Applications of dry chain technology to maintain high seed viability in tropical climates"

_PeerJ, doi:10.7717/peerj.18146_

## Round 0.1 · original submission · Major Revisions

This article provides detailed descriptions of processes and techniques related to dry chain methodology that contribute to its technical depth.
One of the reviewers (Reviewer 1) considers that the work presents deficiencies in its structure and in the clarity of its methods and results and ruled to reject it. However, the other two reviewers have major reservations and comments about your manuscript. Given this, I would like to see a major revision dealing with all the comments. Please revise paying particular attention to the more critical comments, especially in those in the second section, “Original data on the application of the dry chain”, because many inconsistencies are pointed out in the presentation of the national case studies, it is necessary to unify this section with a similar format for all the national examples that are presented. they mention. More clarity is also necessary in the entire description of the methodology used.

Please be aware that we consider these revisions to be major, and your revised manuscript will probably have to be re-reviewed.

If you decide to improve these points, your article will undoubtedly have a good impact and will also contribute significantly to the existing knowledge in this area of research.

Reviewer 1 ·

Basic reporting

Introduction and background of this manuscript is too long, without a clear background where the problem of study and the objectives of the study are defined.
The structure does not follow the format of a scientific text nor that of the PeerJ standards. The manuscript does not include a description of the methodology of any experimental description appropriate to the field. It also does not include a clear results section.
The structure of the manuscript presents as a chapter 1 a broad bibliographic review of seed drying systems without critical analysis, and on chapter 2 information with different approaches and techniques from different countries without a discussion that provides perspectives and offers explanations for the observed data.
They directly address the conclusions of the work without a clear objective.
The figures and tables are irrelevant because they are cases specific to a particular country or method. They do not summarize the possible results obtained that show trends, comparisons or general analyzes of the different countries that were analyzed.

Experimental design

The Research question is not defined. The relevance of the study is not indicated. Relevant. The study gives the impression that it is not a rigorous investigation because it lacks methods and results, consequently, it is not possible to replicate the study.

Validity of the findings

Due to all the shortcomings that the manuscript presents, it is not possible to visualize its possible impact and novelty.
The data underlying the research is not provided because it lacks a results section.
The conclusions are described, but they are not linked to an original research question that is supported by results.

Additional comments

The structure does not follow the format of a scientific text nor that of the PeerJ standards. It is suggested to restructure the manuscript, focusing on the most relevant results that provide important information on the topic.

·

Basic reporting

I commend you on your efforts in addressing the concept of the "dry chain" within the context of the seed area.

As I delved into the article, I found the discussion surrounding the dry chain to be intriguing. However, I believe there is an opportunity to enhance clarity regarding the distinction between the drying process and the dry chain concept. Providing a more explicit explanation of how the dry chain builds upon traditional drying methods and its unique attributes would greatly benefit readers in understanding its significance within the broader context of the seed process.

Furthermore, while reading through the article, I found myself curious about the scale of the dry chain technique implementation. It would be beneficial for readers to have a clearer understanding of the practical application of the dry chain, including details such as the capacity or volume it can accommodate. Specifically, providing insights into the number of seeds or seed bags the dry chain can handle would help readers assess its feasibility and potential impact in real-world scenarios.

Experimental design

As I delved into the article, I noticed that the section discussing unpublished results lacked detailed descriptions regarding the methodology employed. Specifically, there is a lack of clarity regarding how certain procedures were conducted, such as the germination test mentioned as an example. The calculus of how the dryer was adjusted is another. There isn't data about the initial humidity and the final one and how many times was used. The seeds are freshly harvested, if not how were stored. What means high-quality seed? It would be beneficial for readers to understand whether the germination test followed established protocols, such as those outlined by the International Seed Testing Association (ISTA), or if alternative methodologies were utilized.

Providing such details not only enhances the transparency and reproducibility of your research but also allows readers to contextualize the findings within established standards and practices within the field. Additionally, clarity regarding the methodology strengthens the credibility of the results and ensures they can be effectively evaluated and compared with other studies.

I kindly suggest revisiting the unpublished results section to incorporate more comprehensive descriptions of the methodologies employed, including specific references to relevant protocols or guidelines followed, such as ISTA rules for germination testing. This adjustment will greatly enhance the clarity and robustness of your findings, further contributing to the overall impact of your research.

Validity of the findings

As I engaged with the content, I found the section labeled "Original Data" to be intriguing. However, upon closer examination, I observed that this section appears to resemble an extension project rather than a traditional research endeavor. The data and subsequent discussion provided in this section lack adequate integration with the material and methods previously outlined, leading to a disjointed narrative flow.

Additionally, I noted a few specific areas within the "Original Data" section that may benefit from further clarification and discussion. For instance, Figure 3, while visually presented, lacks clear explanatory context, rendering its significance ambiguous to readers. Merely relying on color changes without accompanying textual elucidation may result in confusion and hinder comprehension.

Furthermore, Table 2 is presented without subsequent discussion or analysis within the text, leaving readers without a deeper understanding of its implications or relevance to the broader research context.

To address these concerns and enhance the coherence and clarity of your article, I recommend revisiting the "Original Data" section to ensure a more seamless integration of data, discussion, and interpretation within the framework established in the material and methods. Providing explicit connections between the data presented and the theoretical underpinnings outlined earlier will strengthen the overall narrative and facilitate reader comprehension.

Moreover, I encourage you to revisit Figure 3 to augment its clarity by incorporating descriptive annotations or contextual explanations that elucidate its significance within the context of your research objectives.

I believe that by addressing these points, your article will not only enhance its impact but also contribute significantly to the existing body of knowledge in the field.

·

Basic reporting

The manuscript is generally well-written and provides a thorough and much-needed review of relevant and potentially impactful technology. The abstract and introduction are very clear and easy to follow. The article is divided into two very distinct sections: the review and the data/experiences.
I found the review section excellent, interesting, insightful, and easy to follow. My main suggestion is that the authors divide it into smaller sub-sections with informative thematic titles, as this would help the reader navigate the content (currently uninterrupted 3000 words). Smaller edits are presented in the general comments section.
I found the second section, “Original data on the application of the dry chain”, harder to read. More than “original data,” most of this section presents a variety of “Examples of practical implementation of the dry chain technology across tropical countries.”
There are lots of inconsistencies in the presentation across the national case studies.
I suggest the authors decide what to focus on (project description, social and economic benefit, research, and date) and develop a common template to use for developing each section more coherently. As a reader, it would be much easier to follow. Of all the examples, I personally found the Nepal one better structured and more relevant to this manuscript. It would be great if the other sections could be restructured to a similar format.
The whole Bangladesh section does not present any data but is a summary of the project.
The Costa Rica example should be a stand-alone paper; it’s very long for a sub-section (1400 words) with over-emphasis on the methods (900) and a rushed presentation of results and discussion.
I found the report of the Nepal example the best structured and more in line with the overall aim of the paper. Brief introduction of the project, clear presentation of the results and relevant conclusion. I would suggest restructuring the other sections to be of similar structure, content and length (600 words)
The Pakistan section is similar to the Bangladesh one, with limited data presentation in parts but an extra section that reads more as a review referencing the study with the results.
The South Pacific section presents the project but does not have any data yet.

Experimental design

Being mostly a review this section is not relevant

Validity of the findings

Being mostly a review this section is not relevant

Additional comments

It will be useful in the introduction to specify what short-medium term storage means in terms of months/years.
In the review part, it would be useful to report the time of storage tested for the various examples in lines 180, 191, 209, 214, 217, 235, and 238. I am aware that such data are presented in the supporting information table. However, it is inconvenient for the reader to constantly jump from paper to table, and given the importance of the information, I think it would be better to include it directly in the manuscript. It’ll make for a much better reading experience.
Line 96, “This article aims to provide”
Line 133: remove "also"
Line 117: New paragraph when discussing the second second of the paper
Line 123: remove “indeed”
Line 153-155: I would suggest to rephrase this sentence
Line 156, as written here, it seems that MC and eRH are the same thing, but they are actually different and it might be worth explaining this difference.
Line 159 might be worth briefly mentioning how this card works (e.g, change colour with a change in eRH)
Line 163: remove "latter"
Line 222, consider this reference: Marin, M., Laverack, G., Powell, A.A., Matthews, S., 2018. Potential of the electrical conductivity of seed soak water and early counts of radicle emergence to assess seed quality in some native species. Seed Science and Technology 46, 71–86. https://doi.org/10.15258/sst.2018.46.1.07
Line 244: please rephrase, not clear
Line 282-284: This sentence needs to be supported either by data or a reference. Because of the conflict of interest of one of the authors, who is a supplier of drying beads, this sentence could be interpreted as marketing rather than a factual statement.
Line 289: It would be good if you could cite the paper or website where such a method is described
Line 310: It might be worth mentioning the risk of overheating the seeds if they are too moist and exposed to very charged beads. If I remember correctly, the charged beads heat up considerably when exposed to moisture. It should be advised that seeds are not too wet when exposed to drying beads (and why it’s important to air or sun pre/drying).
Line 330: It might be worth mentioning that further studies are needed to understand the socio-economic implications of the use of dry chain technology on communities.
Line 345: It should be mentioned that a significant portion of native plants from the tropics have recalcitrant seeds, and seed storage behaviour should be understood before dry technology for restoration is implemented.
Line 476: You could reference the Seed Information Database calculators https://ser-sid.org/viability/moisture-equilibrium
Line 693: What are Purdue Improved Crop Storage (PICS) bags?
Line 565: correct kg

Figure 3: It’s very hard to see site type as the tringles overlap with the circles. I would suggest using dotted lines for Plains/Hills

---

## Round 0.2 · Major Revisions

I believe that a good job was done in addressing the points previously suggested by the reviewers. The effort in restructuring the manuscript to improve its quality is notable. However, there are still some issues to be resolved before this manuscript is ready for publication.

The explanation of the differences and relationship between MC and eRH still needs to be improved. Furthermore, the inconsistencies between the experimental case studies should also be considered, and the examples from Mexico and Costa Rica should be rewritten to give them a format similar to that of Nepal.

·

Basic reporting

The authors have done a good job addressing the issues the reviewers raised in the previous submission. They put much effort into restructuring the manuscript to improve its clarity and readability.

Concerning the structure of the manuscript, I acknowledge the point raised by one of the reviewers that it does not follow the standard scientific manuscript structure, being a hybrid between a review and the presentation of case studies. I personally don't mind the "non-hortodox" structure of the paper, and I would not consider that a sufficient reason for rejection, however, I would leave this decision with the editor.

The subdivision of the review section of the manuscript greatly improves the readability and allows the reader to navigate through the key topics covered in the review quickly.
My main recommendation for this section is to improve the explanation of the differences between MC and eRH and how they are related. I think it would be easier for readers unfamiliar with this concept to have it clearly explained at the beginning, so whenever they encounter those terms in the manuscript, they already know what they refer to. There's no need to re-explain it every time.

The subdivision of the case studies between experimental and descriptive is also a good improvement. However, I still believe that the inconsistencies between the experimental case studies still need to be addressed before this manuscript is ready for publication.

As expressed in the previous review, I found the Nepal case study to be better structured and easier to follow. I would recommend that the authors drastically rework the examples from Mexico and Costa Rica to a format similar to Nepal. I appreciate that many methodological details would be lost in abbreviated versions; however, those could be submitted as supplementary material.

Further comments are provided as edits in the submitted pdf file.

Experimental design

The experimental designs presented in the experimental case studies need to be homogenised and simplified.

Validity of the findings

This manuscript has the value of effectively summarising the available knowledge on the drying bead technology in the review section to ensure seed viability in storage and provide useful insight with the experimental and descriptive case studies. Better standardisation in presenting the results of the experimental trials would make it easier to evaluate their robustness. Conclusions are generally well stated.

---

## Round 0.3 · accepted · Accept

Thank you very much for the effort made to improve this manuscript according to the reviewers' comments. After checking this latest version, I consider that your manuscript is ready to be published in PeerJ.